# Liver Brain Interactions: Focus on FGF21 a Systematic Review

**DOI:** 10.3390/ijms232113318

**Published:** 2022-11-01

**Authors:** Eva Prida, Sara Álvarez-Delgado, Raquel Pérez-Lois, Mateo Soto-Tielas, Ana Estany-Gestal, Johan Fernø, Luisa María Seoane, Mar Quiñones, Omar Al-Massadi

**Affiliations:** 1Instituto de Investigación Sanitaria de Santiago de Compostela, Complexo Hospitalario Universitario de Santiago (CHUS/SERGAS), Travesía da Choupana s/n, 15706 Santiago de Compostela, Spain; 2CIBER de Fisiopatología de la Obesidad y la Nutrición, Instituto de Salud Carlos III, 15706 Santiago de Compostela, Spain; 3Unidad de Metodología de la Investigación, Fundación Instituto de Investigación de Santiago (FIDIS), 15706 Santiago de Compostela, Spain; 4Hormone Laboratory, Department of Biochemistry and Pharmacology, Haukeland University Hospital, 5201 Bergen, Norway

**Keywords:** obesity, energy balance, food intake, hypothalamus, FGF21

## Abstract

Fibroblast growth factor 21 is a pleiotropic hormone secreted mainly by the liver in response to metabolic and nutritional challenges. Physiologically, fibroblast growth factor 21 plays a key role in mediating the metabolic responses to fasting or starvation and acts as an important regulator of energy homeostasis, glucose and lipid metabolism, and insulin sensitivity, in part by its direct action on the central nervous system. Accordingly, pharmacological recombinant fibroblast growth factor 21 therapies have been shown to counteract obesity and its related metabolic disorders in both rodents and nonhuman primates. In this systematic review, we discuss how fibroblast growth factor 21 regulates metabolism and its interactions with the central nervous system. In addition, we also state our vision for possible therapeutic uses of this hepatic-brain axis.

## 1. Introduction

### 1.1. Obesity

Changes in diet and physical activity have increased the prevalence of obesity in a relatively short time span [1,2,3,4]. Obesity is defined as an abnormal or excessive accumulation of fat that can be detrimental to health and is caused by excessive nutrient intake over time [5].

The biological mechanisms that regulate food intake were evolutionary adapted in periods where nutrients were scarce and the ability to overeat and store energy when food was present was crucial for survival. This is in stark contrast to our current situation, where food is present in excess in most societies, and our propensity to overeat when food is available, together with the great capacity of our body to store energy in the form of fat, rather represents a great disadvantage to modern humans and gives rise to a range of human health problems [5].

Unlike other common diseases, obesity seems to have an obvious solution that starts with the adjustment of food intake and energy expenditure. However, obesity is a complex and multifactorial disease that is composed of biological, social, and behavioral influences, and treatment by lifestyle changes has been shown to be ineffective [4,6,7]. Therefore, biological and clinical evidence reveals an interaction between genes and the environment that discredits the belief that body weight can be controlled exclusively voluntarily [8].

Obesity is associated with comorbidities such as cardiovascular diseases, type II diabetes mellitus (T2DM), or cancer, among others [9,10]. Therefore, a better understanding of the underlying causes of obesity is urgently required in order to lay the groundwork for the development of new therapeutic strategies [11,12,13].

### 1.2. Brain Homeostatic Mechanisms and Energy Balance

The Central Nervous System (CNS) plays a fundamental role in the control of the metabolic homeostasis of the organism. Since one of the most important functions for survival is to keep us fed and in good nutritional status, the regulation of the metabolism works through a homeostatic system that balances food intake and energy expenditure. It is a complex system, since it is essential for our survival, and food intake is also controlled by the integration of different cognitive, hedonic, and emotional signals that lead to behavioral, autonomic, and endocrine responses [14,15,16,17]. This maintenance of homeostasis is of great biological importance in order to guarantee a perfect balance between nutrients and energy [16,17,18].

The hypothalamus is a region of the brain composed of nuclei interconnected by axonal projections and is the most studied area in terms of the regulation of food intake and body weight [17,18,19,20]. Its anatomical location, at the base of the brain and adjacent to the median eminence (an organ that receives abundant capillary vascularization), allows it to capture the nutrients and hormones secreted into the bloodstream that send information about the energy state, acting as homeostatic feedback signals in order to maintain this metabolic balance [1,8,21].

The hypothalamus, especially the mediobasal hypothalamus, is an important site of action. Included in this neural network are the arcuate nucleus (ARC) and the ventromedial nucleus (VMH). The ARC nucleus is best positioned to receive signals from the periphery and develop a homeostatic response to peripheral tissues, and is thus considered the “master hypothalamic center” for feeding control [8,19].

In the ARC, we can distinguish two differentiated neuronal populations with antagonistic functions. On the one hand, a neuronal population that expresses neuropeptide Y/Agouti-related peptide (NPY/AgRP) induces a positive energy balance; and on the other hand, neurons that express proopiomelanocortin (POMC), which in turn, induces a negative energy balance. Both types of neurons regulate food intake, energy expenditure, and nutrient partitioning [22,23,24,25].

ARC neurons send projections primarily to other “second-order” neurons located in other hypothalamic nuclei, such as the dorsomedial nucleus, paraventricular nucleus (PVH), or lateral hypothalamic area (LHA) [19,20,21]. These nuclei are responsible for integrating hormonal and nutritional metabolic signals from the peripheral circulation, generating a coordinated response [20,22].

Peripheral signals that regulate metabolic control in the CNS may originate from many organs. However, in recent years, the importance of the liver has gained attention and it is considered a master metabolic organ that integrates peripheral nutrient status that is signaled to the brain.

Therefore, in this review, we discuss how a signal derived from the liver, the peptide fibroblast growth factor 21 (FGF21), controls food intake and energy balance and how some of its metabolic actions are induced by its activity in the CNS.

### 1.3. Fibroblast Growth Factor21 and Metabolic Control

FGF21 is a pleiotropic hormone that was discovered in the year 2000 [26]. FGF21 is secreted mainly by the liver in response to metabolic and nutritional challenges [27,28], but is also expressed by adipose tissue, skeletal muscle, and the pancreas [29,30,31,32,33]. FGF21 acts through cell-surface receptors comprised of conventional FGF receptors (FGFRs), with tyrosine kinase activity in complex with the single-pass transmembrane protein β-Klotho. These receptors are relatively abundantly expressed, both in peripheral tissues, such as brown adipose tissue (BAT) and white adipose tissue (WAT), as well as in some regions of the CNS, such as the hypothalamus and hindbrain [34,35,36,37,38,39,40,41,42,43].

FGF21 acts as an important regulator of energy homeostasis, glucose and lipid metabolism, and insulin sensitivity (Figure 1).

Physiologically, FGF21 plays a key role in mediating the metabolic responses to fasting or starvation, including fatty acid oxidation and ketogenesis [32,33,44]. FGF21 in the liver may also be induced by low protein and high carbohydrate diets, and it has broad effects on glucose and fatty acid metabolism [34,45]. Interestingly, while low protein diets increase FGF21 in the liver, it has also been reported that they decrease the concentration in the hypothalamus [46]. Importantly, high levels of FGF21 have been associated with different metabolic diseases, such as obesity [47], T2DM [48], non-alcoholic steatohepatitis (NASH) [49], cardiometabolic disorders [50], and congenital or acquired lipodystrophy [51,52]. Interestingly, FGF21 levels in the muscle are associated with mitochondrial inflammatory myopathy, leading to altered myofiber morphology [53]. FGF21 thus serves as a disease marker, but rather than driving disease development, it is believed that the elevated FGF21 levels are not sufficient to counteract disease development. Consistently, pharmacological recombinant FGF21 therapies have been shown to counteract obesity and its related metabolic disorders in both rodents and nonhuman primates [54,55,56,57,58,59]. Furthermore, FGF21 is the downstream target of both peroxisome proliferator-activated receptor-alpha (PPARa) and gamma (PPARg), and a growing body of evidence suggests that the glucose-lowering and insulin-sensitizing effects of the PPARg agonist thiazolidinediones and the therapeutic benefits of the PPARa agonist fenofibrate on lipid profiles are mediated, in part, by FGF21 induction [60,61,62].

While the liver is the predominant site for FGF21 production, adipocytes are suggested to be the main target of FGF21 action [35,63,64] (Figure 1). FGF21 acts directly on adipocytes to stimulate glucose uptake, and adiponectin secretion [55,65,66,67]. In WAT, FGF21 stimulates glucose uptake in an insulin-independent manner [56], modulates lipolysis [68], and potentiates PPARg activity [61]. There is also compelling evidence showing that FGF21 is involved in the thermogenic functions of brown adipocytes [69,70]. In this sense, FGF21 is expressed and secreted in both WAT and BAT [47,69], and the autocrine actions of FGF21 in adipocytes play obligatory roles in mediating the metabolic benefits of PPARg on glucose homeostasis and peripheral insulin sensitivity by forming a feed-forward loop with this nuclear receptor [61]. Moreover, FGF21 injection increased energy expenditure and adiponectin secretion from adipose tissue. Interestingly, the increased energy expenditure by FGF21 administration was attenuated in adiponectin-null mice [65,66]. The requirement of adiponectin for the full effect of FGF21 suggests that adiponectin is important for fatty acid mobilization in WAT under lipolytic conditions. Notably, lipodystrophic mice with reduced adipose tissue are refractory to both acute and chronic effects of systemic FGF21 administration, which is used to decrease blood glucose and increase insulin sensitivity. FGF21 responsiveness was completely restored after the transplantation of WAT into these mice, confirming that adipose tissue is a predominant site contributing to the antidiabetic activities of FGF21 [63]. However, it is currently unclear how FGF21 controls systemic metabolic homeostasis via its actions in adipocytes.

### 1.4. Central Fibroblast Growth Factor 21 Actions

FGF21 increases energy expenditure and decreases body weight and blood glucose, insulin, and hepatic triglyceride concentrations in rodent models of obesity, in part by acting directly on the nervous system to induce sympathetic outflow to BAT and WAT, promoting in turn thermogenesis and “browning” in these tissues [71,72,73,74] (Figure 2).

In agreement with the pharmacological data, genetic loss of function models of FGF21 aggravates obesity-induced and impairs the thermogenic response, possibly via increased hypothalamic inflammation [75]. Interestingly, by using B-klotho tissue-specific loss-of-function models, the actions of FGF21 on body weight were found to not rely on the ability of FGFs to activate b-klotho in the liver or adipose tissue. In fact, FGF21 required β-klotho-containing receptor complexes in neurons to decrease body weight and circulating glucose and insulin concentrations in diet-induced obese mice [73] (Figure 2).

These data imply that the therapeutic efficacy of FGF-based drugs for treating metabolic diseases relies on their ability to activate b-klotho in the CNS [74]. However, other reports highlight the importance of FGFR1 in the FGF21 action on glucose homeostasis during prolonged fasting. FGF21 acts directly on the hypothalamic neurons to activate the mitogen-activated protein kinase extracellular signal-related kinase 1/2, thereby stimulating the expression of corticotropin-releasing hormone by activation of the transcription factor cAMP response element binding protein in a process dependent on FGFR1 [76].

On the other hand, it has been shown that physiologic, transgenic, and pharmacological activation of FGF21 signaling increases the total caloric intake in rodents [3,56,72,77,78] and fish [79]. Moreover, pharmacological and genetic studies show that FGF21 also induces water intake in part by its actions in the hypothalamic SIM-1 positive neurons [80]. However, as we pointed out before, the genetic or pharmacological activation of FGF21 has the primary effect of increasing thermogenesis and energy expenditure, thereby causing overall weight loss.

In this sense, it is clear that the nervous system plays a crucial role in regulating FGF21’s effect on the intake of liquids or food. FGF21 crosses the blood–brain barrier through simple diffusion [81], and its effects are significantly blunted in animals lacking FGF21-receptors broadly in neurons [78,82,83,84,85].

It is not clear which neuronal mechanisms are behind this orexigenic action of FGF21. Nevertheless, when β-klotho was deleted from vesicular glutamatergic transporter 2 (Vglut2)-expressing neurons, the feeding response to dietary protein was lost [86] (Figure 3). Moreover, FGF21 acts in the ARC to increase the expression of the AgRP and NPY (Figure 3), whereas different studies report about reducing or not affecting cocaine and amphetamine-regulated transcripts or POMC [54,87,88].

Furthermore, some studies show that brain FGF21 signaling is necessary not only for normal food intake but also for macronutrient preference [78,82,83,84]. In this sense, it has been proposed that β-klotho-containing receptor complexes in PVN neurons [82] are part of the neurocircuit mechanisms underlying FGF21’s effect on sweet taste preference (Figure 3). Moreover, deleting β-klotho from Vglut2 neurons, but not gamma-aminobutyric acid (GABA)-ergic or dopaminergic neurons, eliminated the effect of FGF21 on the consumption of sucrose or saccharin [83]. Conversely, genetic activation of β-klotho in Vglut2 neurons decreased sucrose preference [85].

Interestingly, recent studies have shown that FGF21 is produced and secreted in hypothalamic tanycytes [89,90] and activates GABA-containing neurons expressing dopamine receptor 2 (DRd2) in the LHA, that in turn, has been shown to be part of the molecular pathway of the protective effect of prolonged lactation on obesity [90] (Figure 3). Notably, the improvement in body weight due to prolonged breastfeeding in obese mice is due to an increase in the thermogenesis of BAT but not to reduced food intake [90].

In contrast, another study proposes that FGF21 is not expressed in the hypothalamus and, thus, it cannot perform metabolic actions or induce sugar intake in an autocrine manner in this brain area. Instead, it was proposed that FGF21 is exclusively produced from the retrosplenial cortex, where it enhances spatial memory [91].

Nevertheless, accumulating evidence suggests that FGF21 is expressed and mediates its metabolic actions in the brain, and one study links the action of FGF21 and sweet tasting with the mesolimbic dopamine pathway, an area pertaining to the reward system (Figure 4).

Sweets activate ventral tegmental area (VTA) dopamine neurons and increase the dopamine release in the nucleus accumbens (NAc) [87]. Another study supports this neuronal connectivity but with opposite results, and shows that chronic FGF21 administration significantly reduces dopamine and dopamine-related metabolites and increases the expression of the dopamine transporter in the NAc. Moreover, this last study showed that the expression of catechol-O-methyl transferase, an enzyme degrading dopamine, was reduced in the VTA but not in the NAc after FGF21 treatment [84]. These data support the role of FGF21 in modulating dopamine signaling, but more studies are necessary to understand the true relationship between FGF21 and the dopaminergic system.

Interestingly, the FGF21 response to sucrose is associated with body mass index and dorsal striatal signaling in humans, supporting previous evidence in animal models [92].

In conclusion, FGF21 regulates food preference and is proposed to act in areas pretraining the reward system. This fact may open new therapeutic avenues for the use of FGF21 to ameliorate diseases caused by disturbances in homeostatic and hedonic food drives. This assumption is supported by the study of Pena Leon et al. that emphasizes this interconnection by the action of FGF21 on DRd2 in the LHA, an area of the brain that regulates food intake and food reward.

### 1.5. Therapeutic Use of FGF21

Multiple FGF21 analogues have been developed for treating metabolic diseases [93]. The FGF21 analogue LY2405319 was subcutaneously administered for 28 days to patients with obesity and T2DM, resulting in attenuated dyslipidaemia, reduced body weight and plasma insulin, and increased adiponectin levels [94]. Interestingly, this analogue has also been shown to protect against other derangements induced by the metabolic syndrome, such as neurodegeneration [95]. Moreover, the long-acting FGF21 analogue PF-05231023 caused a marked reduction in serum triglycerides but not in body weight when intravenously administered to hypertriglyceridemic obese people, with or without diabetes, who were already treated with atorvastatin [96]. Further, Pegbelfermin (BMS-986036), a PEGylated human FGF21 analogue, ameliorated dyslipidaemia, increased adiponectin levels, and decreased the levels of fibrosis markers in T2DM patients [97]. Consistently, pegbelfermin considerably reduced the hepatic fat fraction, the markers of hepatic injury, and the biomarkers of fibrosis, and ameliorated dyslipidaemia and increased adiponectin levels, without significantly changing body weight, in a phase IIa clinical trial in NASH patients [98]. Another line of research showed that a bispecific antibody named BFKB8488A that activates the FGFR1/B-klotho complex reduces body weight and induces a sustained improvement in cardiometabolic parameters in obese humans. Interestingly, treatment with BFKB8488A also led to a trend towards a reduction in preference for sweet tastes and carbohydrate intake [99]. Moreover, an analogue of FGF19, another FGF family member (NGM282 or Aldafermin), was tested in NASH patients with promising results [100,101,102,103]. Importantly, in a phase 2 trial, Aldafermin was shown to reduce liver fat and generated a trend towards the improvement of hepatic fibrosis in patients suffering from NASH [103]. Finally, it is worth mentioning that a dual agonist composed by GLP-1 and FGF21 has superior efficacy on body weight control and glycemic control compared to a mixture of GLP-1 and FGF21 [104,105].

Accordingly, in the light of these findings and the success of the unimolecular polypharmacy [106] for the development of a novel generation of drugs (e.g., tirzepatide) to treat obesity and diabetes [107], the development of dual agonists that include FGF21 may provide an interesting tool for the treatment of metabolic diseases.

## 2. Methods

### Study Design, Literature Search and Data Collection

Our aim was to compile the most important literature that provides evidence for how FGF21 regulates metabolism, focusing on the interaction between liver and brain. The review was designed according to the Preferred Reporting Items for Systematic Reviews and Meta-Analyses (PRISMA) guidelines [108]. Date restrictions were not applied during the literature search. The literature search was primarily conducted using the PubMed database, as well as the reference lists of the selected studies, only for manuscripts written in English. The titles and abstracts of all electronic articles were screened to assess their eligibility.

Our search was performed using the following keywords/MeSH: [“fibroblast growth factor 21” and “obesity”], [“fibroblast growth factor 21” and “feeding behavior”], [“fibroblast growth factor 21” and “liver”], [“fibroblast growth factor 21” and “brain”], [“fibroblast growth factor 21” and “Diabetes Mellitus, Type 2”], [“fibroblast growth factor 21” and “Non-alcoholic Fatty Liver Disease”], [“fibroblast growth factor 21” and “adipose Tissue, Brown”], [“fibroblast growth factor 21” and “Thermogenesis”], [“fibroblast growth factor 21” and “blood glucose”], [“fibroblast growth factor 21” and “insulin resistance”], [“fibroblast growth factor 21” and “humans”]. Two authors (E.P. and S.A.-D) independently screened all articles for eligibility, while potential disagreements were resolved by consensus among all authors.

We sought human studies, experimental studies, and reviews published in medical journals prior to 29 September 2022, while case reports, editorials, conference abstracts, and posters were excluded. Amongst the 7167 published papers considered, 1421 were excluded after duplicate removal and 5746 were excluded during the screening phase because they were out of the review scope as they did not concern FGF21 or not explicitly assess changes in energy balance or glucose homeostasis.

The remaining 750 full-text papers were assessed for eligibility, and 673 were excluded after abstract or full text screening. Finally, 77 studies were selected, plus 27 identified from their reference lists to the review. The flow chart of the selection process is reported in Figure 5.

## 3. Results and Discussion

The articles referenced in this mini-review focused on general aspects of obesity [1,2,3,4,6,9,10,13,14,21], neuroendocrine control of metabolism [5,7,8,15,16,17,18,19,20,22,23,24,25], non-alcoholic fatty liver disease [11,12], unimolecular polypharmacy [108], and the experimental articles referred to FGF21 expression or referred to the metabolic actions elicited by FGFRs mimetics or activators [26,27,28,29,30,31,32,33,34,35,36,37,38,39,40,41,42,43,44,45,46,47,48,49,50,51,52,53,54,55,56,57,58,59,60,61,62,63,64,65,66,67,68,69,70,71,72,73,74,75,76,77,78,79,80,81,82,83,84,85,86,87,88,89,90,91,92,93,94,95,96,97,98,99,100,101,102,103,104,105,107,108].

Among these studies, only 14 were performed on humans, and only eight of these described an FGF analogue or FGF21 signaling activator. In three of them, the compound reduced body weight [94,99,100], while in the other four, no changes in body weight were observed [96,97,98,101]. In addition to the assessment of body weight, four studies found a reduction in peripheral lipid profile [94,96,97,99] and four found a reduction in liver fat content [97,100,101,103] (Table 1).

These variations could be due to the different nature of the compounds, different doses used, routes of administration, or the heterogeneity of the metabolic state of the patients. All together, we can conclude that FGF21 analogues display an a priori high applicability; however, although compounds based on FGF21 seem to be promising targets against metabolic diseases such as T2DM and NASH, their effect on body weight is still suboptimal. 

The quality assessment was performed by the Cochrane risk-of-bias tool for randomized trial RoB scale in the manuscript regarding only clinical trials (Appendix A), while the outcomes of the qualitative synthesis are presented in Table 2.

## 4. Concluding Remarks

In this review, we covered the metabolic actions of FGF21 and its effect on energy balance. This hepatic signal appears not only as a key metabolic factor directly implicated in T2DM and non-alcoholic fatty liver disease (NAFLD)/NASH by its action in glucose and lipid metabolism but also as regulators of food intake and body weight through its action on the CNS. Regardless of the great expectations for the use of FGF21 as a therapeutic target, there are differences in the physiological functions of FGF21 between mice and humans, a fact that complicates the translational value of the peptide [109].

Therefore, in spite of the recent advances in defining the intracellular function and regulation of FGF21, the molecular mechanisms mediating its actions and its physiological and pathophysiological roles in CNS homeostasis are not totally understood.

In conclusion, FGF21 continues to emerge as an interesting tool for the study and precise understanding of metabolic and liver diseases.

## Figures and Tables

**Figure 1 ijms-23-13318-f001:**
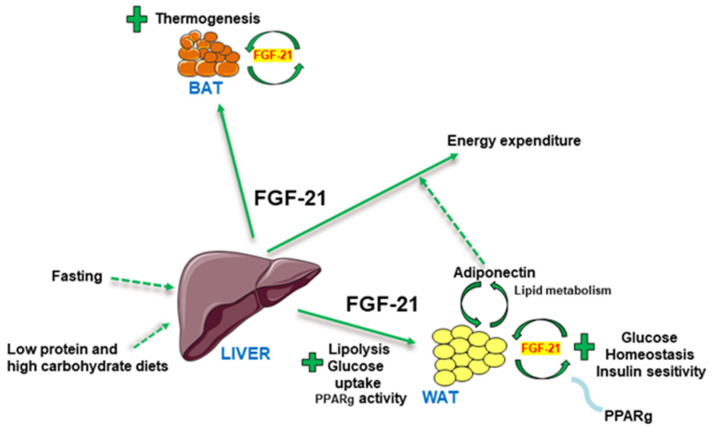
Description of the peripheral actions elicited by FGF21. FGF21 is a hepatic signal that improves several metabolic parameters such as energy expenditure, BAT thermogenesis, lipolysis, and glucose homeostasis. Abbreviations used: BAT: brown adipose tissue; FGF21: fibroblast growth factor-21; PPARg: Peroxisome proliferator-activated receptor gamma; WAT: white adipose tissue.

**Figure 2 ijms-23-13318-f002:**
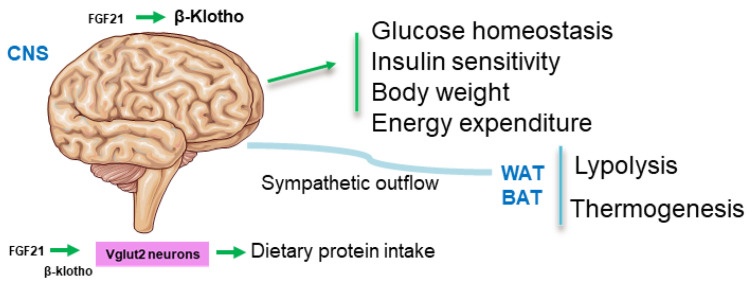
FGF21/β-Klotho interactions in the CNS in metabolic control: FGF21 required β-klotho-containing receptor complexes in neurons to decrease body weight and circulating glucose and insulin concentrations, to induce energy expenditure and to signal WAT and BAT by SNS activation. Abbreviations used: CNS: central nervous system; BAT: brown adipose tissue; FGF21: fibroblast growth factor-21; Vglut2: vesicular glutamate transporter 2; WAT: white adipose tissue.

**Figure 3 ijms-23-13318-f003:**
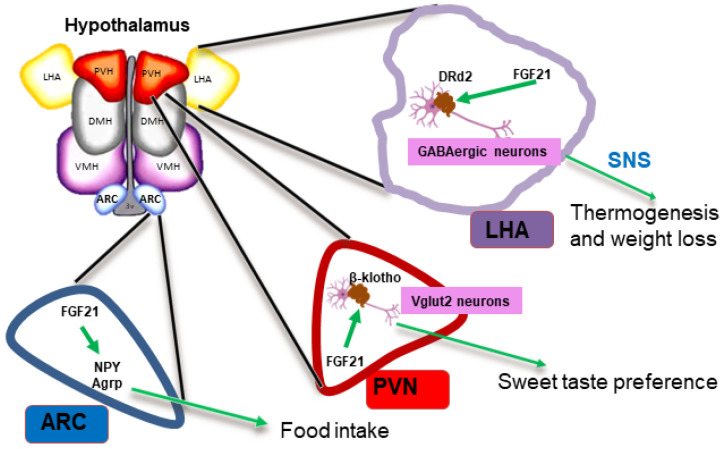
Description of the metabolic actions elicited by FGF21 in the hypothalamus. FGF21. FGF21 is a hepatic signal that induces food intake and improves several metabolic parameters such as energy expenditure, BAT thermogenesis, mainly by its action in the hypothalamus. CNS. Abbreviations used: Agrp: agouti related peptide; ARC: arcuate nucleus; GABA: gamma aminobutyric acid; DMH: dorsomedial hypothalamus; FGF21: fibroblast growth factor-21; LHA: lateral hypothalamic area; PVN: paraventricular hypothalamus; NPY: neuropeptide Y; Vglut2: vesicular glutamate transporter 2; VMH: ventromedial hypothalamus; SNS: sympathetic nervous system.

**Figure 4 ijms-23-13318-f004:**
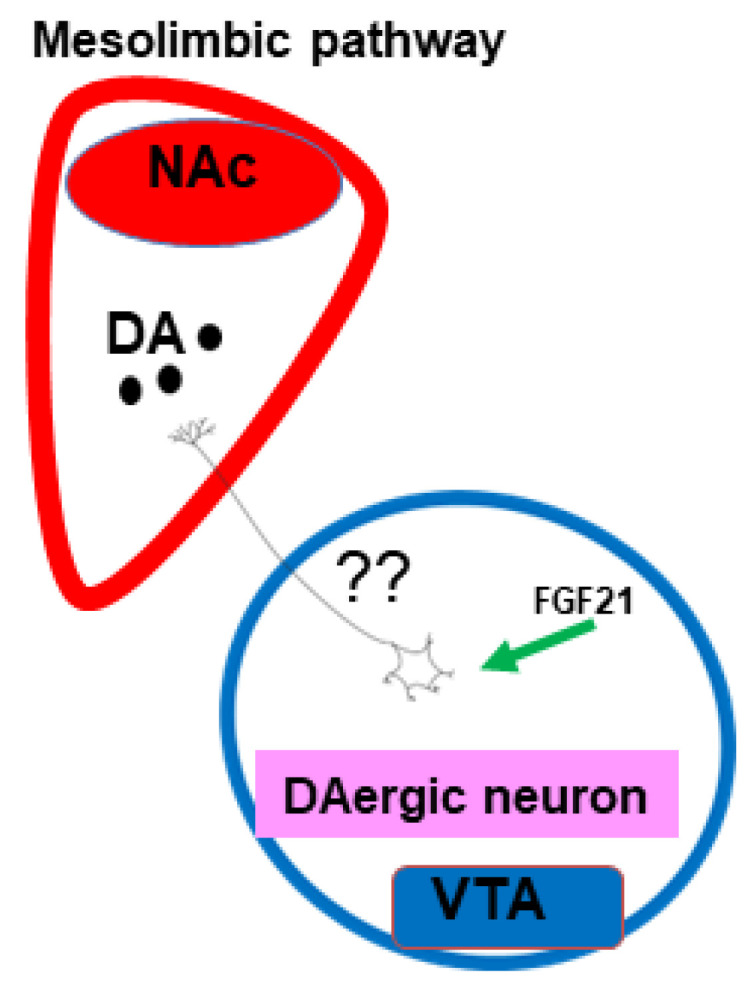
Implications of FGF21 in the mesolimbic system: Abbreviations used: DA: dopamine; NAc: nucleus accumbens. FGF21: fibroblast growth factor-21; VTA: ventral tegmental area.

**Figure 5 ijms-23-13318-f005:**
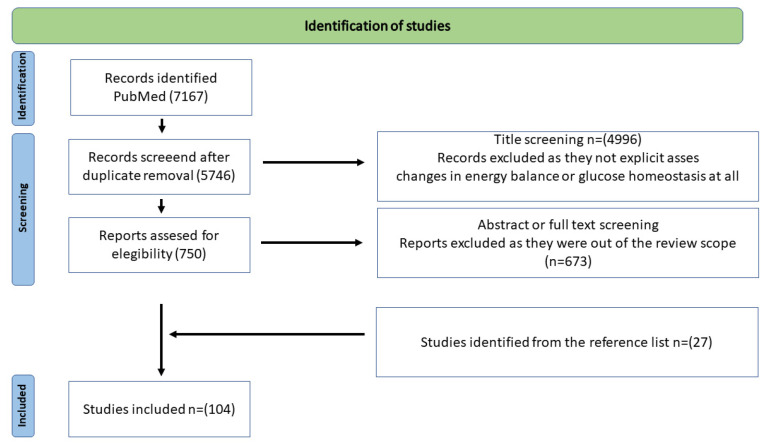
Flowchart of the study. ([“fibroblast growth factor 21” and “obesity”], [“fibroblast growth factor 21” and “feeding behavior”], [“fibroblast growth factor 21” and “liver”], [“fibroblast growth factor 21” and “brain”], [“fibroblast growth factor 21” and “Diabetes Mellitus, Type 2”], [“fibroblast growth factor 21” and “Non-alcoholic Fatty Liver Disease”], [“fibroblast growth factor 21” and “adipose Tissue, Brown”], [“fibroblast growth factor 21” and “Thermogenesis”], [“fibroblast growth factor 21” and “blood glucose”], [“fibroblast growth factor 21” and “insulin resistance”], [“fibroblast growth factor 21” and “humans”]).

**Table 1 ijms-23-13318-t001:** Metabolic actions of FGF21 mimetics or FGF21 signaling activation compounds: BW: body weight; db/db mice: mice lacking leptin receptor; DIO: diet induced obesity; EE: energy expenditure; FGF21: Fibroblast growth hormone 21; GLP-1: Glucagon like peptide-1; ICV: intracerebroventricular; ip: intraperitoneal; iv: intravenous; NAFLD: non-alcoholic fatty liver disease; NASH: non-alcoholic steatohepatitis; ob/ob mice: mice deficient in leptin; sc: subcutaneous; TG: triglyceride.

Study	Specie	Administration	Compound	Effect
2020; Gilroy [104]	Wild type mice	Peripheral	GLP-1 and FGF21 dual agonist	Potently reduces BW Decrease fasting glucose Improves NASH
2021; Pan [105]	Ob/ob mice Db/db mice	Peripheral SC	GLP-1 and FGF21 dual agonist	Potently reduces BW Improves glucose Improve NAFLD
2008; Coskun [54]	Ob/ob and DIO mice	Peripheral IV	recombinant human FGF21	Potently reduces BW Increases EE Improves glucose Improves NAFLD
2005; Kharitonenkov [55]	Ob/ob and db/db mice	Peripheral SC	human FGF21	Reduces plasma glucose and TG
2009; Xu [57]	Ob/ob and DIO mice	Peripheral IP	recombinant human FGF21	Improves glucose tolerance and insulin sensitivity
2007; Kharitonenkov [58]	Diabetic non human primates	Peripheral IV SC	recombinant human FGF21	Mild reduction of BW Reduces plasma glucose and insulin and improves lipid profile
2013; Adams [59]	DIO Mice	Peripheral SC	FGF21 analogue, LY2405319	Reduction of BW Reduces plasma glucose and insulin and improves lipid profile
2013; Holland [66]	Ob/ob and DIO mice	Peripheral SC	recombinant murine FGF21	Reduces plasma glucose and improves insulin sensitivity Increases adiponectin and EE
2015; Douris [71]	WT mice	ICV	recombinant murine FGF21	Energy expenditure, thermogenesis, and “browning”
2014; Owen [72]	WT Mice	ICV	recombinant murine FGF21	Increases thermogenesis
2017; Lan [73]	DIO mice	Peripheral IP	FGF21 mimetic antibody	Reduces BW, plasma glucose and insulin, and improves lipid profile
2022: Pena-Leon, [90]	Rats	ICV	recombinant human FGF21	Reduces BW Increases thermogenesis
2013; Gaich [94]	Obese and diabetic Humans	Peripheral SC	FGF21 analog LY2405319	Reduces BW and plasma insulin and improves lipid profile
2017; Kim [96]	Obese humans	Peripheral IV	PF-05231023	Not changes in BW Potently reduces TG
2019; Charles [97]	Obese and diabetic humans	Peripheral SC	Pegbelfermin (BMS-986036), PEGylated FGF21	Non changes in BW Decrease fasting glucose and insulin sensitivity Improves lipid profile Improve fibrosis markers
2019; Sanyal [98]	overweight or obese with NASH	Peripheral SC	Pegbelfermin (BMS-986036), PEGylated FGF21	Non changes in BW Improves NASH
2020; Baruch [99]	Non human primates	Peripheral IV	bispecific anti-FGFR1/KLB agonist antibody BFKB8488A	Reduces BW Reduction in preference for sweet taste and carbohydrate intake
2020; Baruch [99]	overweight or obese	Peripheral SC	bispecific anti-FGFR1/KLB agonist antibody BFKB8488A	Transient reduction in BW Improve lipid profile Decrease fasting insulin
2018;Harrison [100]	NASH patients	Peripheral S.C.	FGF19 analogue, NGM282	Reduces BW Improves NASH
2020;Harrison [101]	NASH patients	Peripheral S.C.	FGF19 analogue, NGM282	Improves NASH
2021;Harrison [103]	NASH patients	Peripheral S.C.	FGF19 analogue NGM282	No changes in BW Reduced liver fat
2015; Talukdar [84]	DIO mice Monkeys	Peripheral S.C. I.V.	recombinant human FGF21 PF-05231023	Supresses sweet preferences
2016; Von Holstein-Rathlou [82]	Mice	Peripheral IP and SC	recombinant human FGF21	Suppresses sugar intake and sweet taste preferences

**Table 2 ijms-23-13318-t002:** Quality synthesis: ANCOVA: analysis of covariance; ALT: alanine aminotransferase; AST: aspartate aminotransferase; BW: body weight; FGF21:Fibroblast growth hormone 21; HDL: high density lipoprotein; LDL: low density lipoprotein; LS: Least squares; NASH: non-alcoholic steatohepatitis; SE: standard errors; SD: mean difference mean; TG: triglycerides.

Study	Model	Data	Compound
2013; Gaich [94]	Mixed effect linear model	BW: LS mean change from baseline = −1.75 (0.65) *p* < 0.05	LY2405319
2017; Kim [96]	Mixed-effects model for repeated measurements	TG: Placebo-adjusted least squares mean 90% CI −62.1, −24.6	PF-05231023
2019; Charles [97]	Longitudinal repeated- measures analysis model	Glucose: LS mean Estimates SE, and two-sided 90% confidence intervals (90% CI −3.46 to 0.22) Fibrosis markers: % of change compared with baseline ALT—20%; AST—8%	BMS-986036
2020; Baruch [99]	Descriptive statistics.	Mean and SD % change for baseline: BW: 2.30 (0.25) Mean % change for baseline: Cardiometabolic parameters: TG: −66% HDL: +34% LDL: −37% Adiponectin: +250%	BFKB8488A
2019; Sanyal [98]	Longitudinal repeated measures analysis and unstructured covariance matrix were used to represent the correlation of the repeated measures within each patient	Hepatic fat: Adjusted mean absolute change compared with placebo Mean and SE, and two-sided 90% confidence intervals (−6.8% vs. −1.3%; *p* = 0.0004)	BMS-986036
2018; Harrison [100]	ANCOVA	BW: LS SD, SE (95% CI), −2.0 (0.9; 3.7 to −0.3) *p* = 0.023 Hepatic fat: LS means With SE, 95% (−11.1 1.4, −13.9 to −8.3; *p* < 0.0001)	NGM282 (FGF19 analogue)
2020; Harrison [101]	Sensitivity analysis Wilcoxon matched pairs signed rank test	Fibrosis scores: change compared with baseline (0.5; 0.9 to −0) *p* = 0.035	NGM282 (FGF19 analogue)
2021; Harrison [103]	ANCOVA	Least squares (LS) Means with standard errors (SE), 95% (difference, −5.0%; 95% CI, −8.0% to −1.9%; *p* = 0.002)	FGF21 analogue aldafermin

## Data Availability

Not applicable.

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
