# Peer review of "Liver Brain Interactions: Focus on FGF21 a Systematic Review"

_ijms, 2022, doi:10.3390/ijms232113318_

Round 1

Reviewer 1 Report

1)     L1nes 108-109/ Figure 1 legend - GH: growth hormone; IGF-1: insulin growth factor-1; NEFAs: non-esterified fatty acids. These abbreviations are not present on the figure.

On Figure 1, the word LIVER is hardly visible. It should be moved outside of the liver picture, or color of the letters should be changed.

2)     From the Figure it is not clear what is the main source of FGF21 in the periphery. Text suggests that it should be liver, but in the Figure only thin arrow points from the liver to the brain. Also, this single arrow depicting liver-derived FGF21 affecting the CNS. But in text we see that it has essential function when signals in WAT and BAT. Adiponectin is shown as simply being secreted from WAT, while its positive feedback loop in this tissue is missing. The figure should be changed to fit the text, missing details, participating factors and links have to be added as well.

3)     The complexity of information concerning effects of FGF21 signaling in the CNS, lack of information if CNS-derived FGF21 has paracrine, or mostly exocrine action, make the understanding of relations between liver and CNS as the main producer of FGF21 confusing. I would even suggest to split figure one to two separate figures. The first one should demonstrate effects of FGF21 in the periphery with a focus on liver. And the second one could concentrate on CNS-derived FGF21, effects of this peptide on particular groups of neurons, and physiological consequences of FGF21 signaling ablation in the CNS.

4)     It would be of much help to add a table with the list of FGF21/FGF21 signaling related drugs (mimics and analogues, agonists, antagonists, blockers, bi-specific antibodies and so on), major effects they induce in mice and humans upon administration, and references on the literature. This will simplify the analysis of presented information by the readers.

5)     Finally, there are several typos and grammar mistakes over the text. Please, take time to edit the manuscript.     

The paper requires significant revision prior to possible acceptance.

Reviewer 2 Report

Presented review paper focuses on the possible connectivity between liver and brain via FGF21 action. The discussed topic is an interesting and novel approach to connect two major organs of the body in terms of looping feedback. In general, I have two very major comments, that must be addressed and fixed before processing further. Please follow my two points.

Firstly, the aim of the review paper is to discuss in the comprehensive manner the recent advances in the discussed area. Here, I see most references dated 2000-2010. Recently, basic databases like W-O-S or PubMed shows over 300 records for FGF21 biology in liver/brain and similar searches. Only few are included in the review. Moreover, there are very well published paper describing the same topic with very similar findings. Please follow: e.g. https://doi.org/10.2337/db14-0541 ; doi.org/10.3892/br.2017.890

https://pubmed.ncbi.nlm.nih.gov/25024372/ ; https://www.nature.com/articles/s41574-020-0396-y and many more. The novelty and very narrow literature search is the major problem that must be fixed.

Moving further, there is a very limited amount of information given regarding the following PRISMA guidelines and description of the searching strategy, which is so crucial for review papers. The Author should include at least: Data sources and searches, Study eligibility criteria, Study selection process, Data extraction, and study quality assessment (assessing the risk of bias (ROB) for each included study), Data synthesis. MeSH terms (in addition/replacement of keywords) are necessary to be included. For each step, it is necessary to explain to the reader with pictures or tables. It is necessary to explain what was drawn at each step to lead to the result. Moreover, a figure showing the PRISMA-based workflow must be drawn accordingly to the Prisma schema. After that, a discussion is valuable even for narrative papers. Description of Data Mining strategy should also be included.

There are some minor grammatical error / minor typos - please go over it carefully.

Please shorten the Conclusions section - it should be a concise summary of the discussed area.

Round 2

Reviewer 2 Report

The Authors addressed all my majors and most of minors in a satisfactory way.

Right now, I feel that the paper is a solid contribution to the field and is fully publishable. However, to make the review as comprehensive as possible, I would strongly suggest including some fresh paper describing the role of FGF21 in metabolism in different diseases, please follow:

https://doi.org/10.3390/jcm8122206

doi: 10.1210/er.2017-00016

doi: 10.15252/embj.201796553

Author Response

"Please see the attachment".
